# Inflammation and Prostate Cancer: Pathological Analysis from Pros-IT CNR 2

**DOI:** 10.3390/cancers15030630

**Published:** 2023-01-19

**Authors:** Francesco Sessa, Rossella Nicoletti, Cosimo De Nunzio, Angelo Porreca, Stefano Maria Magrini, Vincenzo Mirone, Andrea Tubaro, Sergio Serni, Paolo Gontero, Marianna Noale, Stefania Maggi, Mauro Gacci

**Affiliations:** 1Unit of Urological Robotic Surgery and Renal Transplantation, Careggi Hospital, University of Florence, 50134 Florence, Italy; 2Urology Unit, Sant’Andrea Hospital, University of Rome “La Sapienza”, 00189 Rome, Italy; 3Department of Urology, Veneto Institute of Oncology (IOV) IRCCS, 35128 Padua, Italy; 4Radiation Oncology Department, University and Spedali Civili Hospital, 25123 Brescia, Italy; 5Department of Urology, University Federico II, 80138 Naples, Italy; 6Division of Urology, Molinette Hospital, School of Medicine, University of Torino, 10124 Torino, Italy; 7Aging Branch, Neuroscience Institute, National Research Council, 90129 Padua, Italy

**Keywords:** prostate cancer, prostate inflammation, prostate biopsy

## Abstract

**Simple Summary:**

The association between inflammation and cancer is well demonstrated. The aim of our study was to investigate the topographic and quantitative association between inflammation, proliferative inflammatory atrophy and Prostate Cancer (PCa) in frustules from prostate biopsies. We demonstrated an association between prostate inflammation (PI) and PCa in tissue from prostate biopsies, confirming the basis of future research aimed at evaluating anti-inflammatory therapies as a way to prevent PCa.

**Abstract:**

Background: Extensive research effort has been devoted to investigating the link between inflammation and PCa. However, this relationship remains unclear and controversial. The aim of our multi-center study was to investigate this association by histologically evaluating the distribution of PI and PCA in prostate biopsy cores from patients of eight referral centers in Italy. Results: We evaluated 2220 cores from 197 patients; all the frustules were re-evaluated by dedicated pathologists retrospectively. Pathologists assigned IRANI scores and determined the positions of PIs; pathologists also re-evaluated the presence of PCa and relative ISUP grade. PCa was recorded in 749/2220 (33.7%). We divided this sample into a PCa PI group (634/749 cores [84.7%]) and a non-PCa + PI group (1157/1471 cores [78.7%]). We observed a statistically significant difference in the presence of inflammation among cores with cancer (*p* < 0.01). Moreover, periglandular inflammation was higher in the cores with neoplasia, while stromal inflammation was higher in cores without neoplasia (38.5% vs. 31.1% and 55.4% vs. 63.5% *p* < 0.01). Conclusions: In our experience, there is evidence of an association between PI and PCa at a tissue level. Further studies are needed to confirm our findings and to identify patients who might benefit from target therapies to prevent PCa occurrence and/or progression.

## 1. Introduction

Prostate cancer (PCa) is the most common non-cutaneous malignancy diagnosed in men, and its risk factors include advanced age, family history and African American ancestry [1]. Moreover, the pathogenesis of PCa involves environmental factors in addition to hereditary factors. One such potential factor, which has gained a great deal of recent attention in the last years, is the presence of chronic inflammation (PI) [2]. Indeed, the abundance of clinical, molecular and histopathological evidence linking PCa and inflammatory competition has recently caused an increasing interest in the tumor-promoting effects of the inflammatory microenvironment.

PI represents a crucial process leading to a cascade of chemical events or tissue injury for Benign Prostatic Hyperplasia (BPH) pathogenesis and progression (i.e., the damage caused by inflammatory infiltrate leads to tissue disruption and to a chronic process of wound healing that subsequently determines prostatic enlargement) [3,4,5,6]. Thus in recent years, extensive research effort has been devoted to investigating the link between inflammation and PCa, and to finding which inflammatory lesions of the glandular epithelium might act as early forerunners of neoplastic transformation [7,8,9,10,11].

However, this relationship remains unclear and controversial [2]. What is undoubted is that inflammatory processes may lead to a cascade of chemical events that retard the eradication of pathogens, clearing tissue and cellular debris and regenerating the epithelium, as well as the remodeling of the stroma [12]. Therefore, sustained inflammation could generate numerous reactive nitrogen and oxygen species, cytokines, chemokines, and growth factors. Therefore persistent high levels of these factors potentially lead to uncontrolled cellular proliferation and enhanced genomic instability. Thus, the inflammatory state may initiate or promote neoplastic progression if it is able to transform cells present in the local environment into a fully malignant phenotype [12].

Although the notion that immune response localized to the tumor inhibits cancer growth is controversial, it is now clear that some types of tumor-associated immune cells exert actions at some point in PCa’s natural history [13]. In fact, the pivotal role of the immune system in PCa biology has been underlined by the recent development of immunotherapy and vaccines against PCa [11]. In this regard, the quantification of the presence, as well as the location and extent, of immune and inflammatory cells in prostatic tissue might play a role in a new era of targeted and tailored therapies and might be used by clinicians while making therapeutic decisions regarding PCa.

The aim of this study was to investigate the topographic and quantitative relationships among inflammation, proliferative inflammatory atrophy, and PCa.

## 2. Materials and Methods

### 2.1. Study Population

Pros-IT2 is a satellite study of the PROState cancer monitoring in the ITaly project from the National Research Council (Pros-IT CNR). Pros-IT2 is a contemporary, longitudinal and observational study, the aim of which was to monitor the quality of life in PCa patients [14,15]. A non-probability convenience sample of 1705 treatment-naïve patients with histologically confirmed PCa was enrolled in 97 Urology, Radiation Oncology and Medical Oncology facilities located throughout Italy from September 2014 to September 2015 and followed for 60 months from the diagnosis. The present study actively involved the pathology andanatomy departments of 8 centers of the Pros-IT working group, identified considering the number of patients enrolled for the purpose of the main study [Figure 1].

From September 2017, the records of patients from 8 Italian referral centres, who underwent prostate biopsies (PB) for suspected prostate cancer, were retrospectively evaluated. Indications included: elevated PSA levels, abnormal DRE and abnormal imaging findings on MRI or TRUS of the prostate gland when they were performed before PB. Exclusion criteria were: suspicion of clinically acute prostatic inflammation, repeat biopsy, active surveillance, previous BPH surgery, medical treatment with 5α- reductase inhibitors, or saturation biopsies.

The Pros-IT CNR protocol and the amendment for the Pros-IT2 study were approved by the Ethics Committee of the clinical coordinating center located at the Sant’Anna Hospital (Como, Italy) and by the Ethics Committees of each participating center. The study was carried out in accordance with the principles of the Declaration of Helsinki; all patients signed an informed-consent form for data collection.

### 2.2. Patient’s Features

Age (years), body mass index (BMI, kg/m^2^), smoking status, cardiovascular status and history of diabetes were evaluated for each patient within the Pros-IT CNR main study. PSA (ng/mL) was measured using immuno-radiometric tests. The total prostate volume (TPV) and the transition zone volume (TZV) were measured by TRUS using the formula for an ellipsoid (length × width × height × 0.52). All biopsy cores were taken using the standard TRUS transperineal or transrectal technique. The prostates were divided into 14 zones, from the base to the apex, as described in Appendix A. Each core was coded, and it was possible to assign to each of them a specific zone of the prostate, according to Appendix A. Analysis of target cores was excluded in order to avoid skewing phenomena. We did not consider PSA density (ratio between PSA and TPV) in order to avoid conflicting analysis.

### 2.3. Pathological Features

Each core was evaluated by blinded dedicated pathologists who systematically assessed the following features: length (mm); biopsy Gleason Group (BGG) and tumour grade group system (according to the International Society of Urologic Pathology—ISUP) [16]; percentage of cancer that involves each core; biopsy positive (cancer) cores (BPC); prostatic intraepithelial neoplasia (PIN); and chronic inflammatory infiltrate or PCI (defined as the presence of lymphocytes, macrophages and dendritic cells or mixed cells, glandular atrophy, and atypical small acinar cell proliferation).

In each core, we evaluated the presence or absence of chronic inflammation according to these parameters: anatomical location (stromal, periglandular and glandular), grade of inflammation (soft, moderate, severe), the extent of inflammatory infiltrates [focal (<10%), multifocal (10–50%), diffuse (>50%)]. An inflammatory score (IRANI score) [17] (IS) was calculated in accordance with the parameters below, ranging from soft (IS 3–4), moderate (5–6), to severe (7–9). The destruction of the glandular epithelium caused by the massive inflammatory infiltration was considered an additional indicator of inflammation and recorded in the cards as “Glandular disruption” (GD, present or absent). The presence of the amylaceous corpora in the biopsy samples (CA, present or absent) was also noted. For each core, a dedicated form was filled properly (Appendix A).

### 2.4. Statistical Analysis

The collected data were summarized by means and standard deviations (SD), median and interquartile intervals (Q1 = quartile 1; Q3 = quartile 3) for quantitative variables, and frequency distributions for categorical variables. The associations among specific parameters of prostate inflammation (prevailing anatomical localization, degree of inflammation and extent of inflammatory infiltrates), prostate zone with inflammation and the presence of neoplasia for each core were evaluated considering the χ2 test.

*p*-values < 0.05 were considered statistically significant. The analyses were performed using the SAS statistical package, release 9.4 (SAS Institute Inc., Cary, NC, USA).

## 3. Results

Overall, 197 patients fulfilled the inclusion criteria and were included in the Pros-IT2 analysis. For each patient, an average of 11 ± 5 cores (median value: 12) were evaluated, for a total of 2220 frustules.

### 3.1. Baseline Characteristics of Patients

The overall characteristics of the patients included in the study are reported in Table 1. The mean age at diagnosis of PCa was 66 years (65.5 ± 7.5), while the mean BMI was 27 kg/m^2^ (26.7 ± 3.2). Among the patients included, 14% were smokers at diagnosis, while 10% suffered from diabetes requiring oral antidiabetic therapy. Preoperative median PSA was 6.7 ng/mL (Q1 = 4.9; Q3 = 9.8), rectal exploration was negative in 63.1%, clinical T stage was cT1 or cT2 in 96.4%, while 3.6% were cT3 or cT4.

### 3.2. Characteristics of Cores among Patients with PCa

Overall characteristics of the cores among patients with PCa are reported in Table 2.

Prostate cancer was recorded in 749/2220 cores (33.7%). The mean (SD) number of cores of each patient was 11.3 (±4.9), while the mean number of positive cores for each patient was 4.1 (±2.9). Of 749 positive cores, 36.9%, 27.1%, 10.8%, 17.1% and 8.1% were Grade Group 1, 2, 3, 4, and 5, respectively. Considering the risk category defined by the EAU Guidelines (1), 37.4%, 37.4% and 25.2% of the cores with neoplasia were low, intermediate and high risk, respectively.

### 3.3. The Relationship between Inflammation and Neoplasm for Each Core

The relationship between inflammation and neoplasm is summarized in Table 3. The overall cores’ distribution between groups was as follows: PCa + PI group (concomitant presence of cores with inflammation and neoplasia) 634/749 cores (84.7%) and non-PCa + PI group (presence of cores with inflammation without the presence of neoplasia) 1157/1471 cores (78.7%). Proportionally, the number of cores with neoplasia and PI (*n* = 634/749) was higher than the cores with PI and absence of neoplasia (*n* = 1157/1471) (84.7% vs. 78.7%, *p* < 0.01). Moreover, considering the distribution of inflammation, periglandular inflammation was higher in the cores with neoplasia, while stromal inflammation was higher in the cores without neoplasia (38.5% vs. 31.1% and 55.4% vs. 63.5% *p* < 0.01).

Similarly, a moderate grade of inflammation was recorded in 29.5% of cores with neoplasia and in 21.3% of cores without (*p* < 0.01). Nonetheless, a multifocal distribution of inflammatory infiltrates was recorded in 22% of the cores with neoplasia vs. 16.9% of the cores without (*p* = 0.01).

### 3.4. Follow up and Risk of BCR

Considering Pros-IT patients for whom assessment on cores and inflammation were available (*n* = 197), no significant association between inflammation and relapses was found (16.7% among patients without inflammation vs. 14.2% among patients with inflammation; *p* = 0.7281). The percentage of patients with PSA > 0.07 ng/mL at the 12-month follow-up was 12.1% among those with inflammation, 8.3% among those without inflammation (*p* = 0.7017), and at the 24-month follow-up, 13.2% and 0.0%, respectively (*p* = 0.3847).

## 4. Discussion

The purpose of our study was to evaluate the potential association between PI and PCa and to explore the anatomic distribution and the grade of inflammation in the prostatic tissue of two cohorts of patients with and without PCa. Our hypothesis arises from the need to investigate the role that chronic inflammation may have regarding the onset and progression of human cancer via modification of the extracellular matrix and via epithelial–mesenchymal transition.

Chronic inflammation is a common finding within prostatic tissue. The REduction by DUtasteride of PCa Events (REDUCE) trial demonstrated that 77.6% of all prostate biopsies exhibited inflammatory tissue and, among these, the majority (89%) had mild chronic inflammation [18]. Evidence regarding prostatic inflammatory tissue has been gathered not only from prostate biopsies but also from radical prostatectomy specimens and tissue that was removed during transurethal resection for BPH [19]. The inflammatory infiltrates primarily consist of T lymphocytes, macrophages and, less frequently, plasma cells and eosinophils [20].

For many years, chronic inflammation has been suspected of playing a major role in the pathogenesis of cancer [21]. However, this relationship remains unclear and controversial with respect to PCa.

It has been hypothesized that chronic prostatic inflammation causes an immunologic response, promoting tissue damage and subsequent repair, processes that may lead to the enlargement and cancer vulnerability of the gland. A meta-analysis conducted on 11 case-control studies published between 1971 and 1996 (Dennis and co-workers) showed an increased risk of PCa among men with a history of prostatitis [22]. In addition, the first study by Sutcliffe et al. prospectively assessed a correlation between prostatitis and PCa and failed to demonstrate any statistical connection, although a positive association was observed among younger men screened for PCa [23]. Multiple different aetiological agents are thought to contribute to the initiation of prostatic inflammation, including infections, dietary factors, corpora amylacea (and associated physical trauma), hormonal changes and urine reflux [20].

Given the return of common interest in the tumor-promoting effects of the inflammatory microenvironment in the last few years, as well as the potential application of therapeutic targeting approaches combined with classical therapeutic sources, our study aims to fill a priority gap in the literature by evaluating the potential association between prostate inflammation and prostate cancer and explore the anatomic distribution and the grade of inflammation in the prostatic tissue in two cohorts of patients with and without PCa. In this regard, our study provides several key findings to contextualize the relationship between PI and PCa. The first key finding is that a significant distribution of inflammation was recorded within the prostatic tissue. These results reinforce the concept that inflammation promotes tumor cell proliferation, epithelium–mesenchymal transition, angiogenesis and metastasis. The presence of a periglandular infiltrate more represented than the stromal could indicate that the creation of a microenvironment rich in periglandular inflammatory chemokines can dysregulate the growth, and therefore the proliferation, of prostatic acinus cells, acting as a trigger for carcinogenesis. Miller et al. showed that the amount of CD4 + CD25 high regulatory T cells (Treg) is markedly augmented in both tumor tissue and peripheral blood of patients with early-stage PCa [24]. Cell transformation, mediated by transcription factor NF-kB activation that leads to the upregulation of inflammatory cytokines such as TNF-α and IL-6, may be promoted by procarcinogenic inflammatory processes [25]. A second key finding is that the distribution of PI among cores with prostatic neoplasia was mainly in the periglandular zone. From this point of view, our study would seem to support the hypothesis of the creation of an inflammatory periglandular trigger microenvironment. This concept was previously reported by Vral et al. [7] in their study taking up the “injury and regeneration” model for prostate carcinogenesis, a model theorized by Sfanos et al. [20], who showed that the lesions caused by pathogens or pro-inflammatory cytotoxic agents would activate the proliferation of prostate glandular cells, leading to the appearance of epithelial lesions called “Proliferative inflammatory atrophy (PIA)”. Vral et al.’s results indicate a positive association between tissue inflammation, clinical prostatitis and the presumed risk of PIA injury. In this regard, future studies should investigate which role might have an early identifier and the location of inflammation to target an early therapy to limit angiogenetic and mesenchymal transformations. Another key finding was that we adopted a detailed and validated model (the IRANI score) as a benchmark of prostatic inflammation. Although the three most frequent localizations of prostatic inflammation (i.e., glandular, periglandular and stromal) are generally recognized, no consensus has been reached concerning the quantification of the extent and grade of inflammation, nor has any consistent correlation emerged among clinical syndromes or symptoms. Various classification systems were published between 1991 and 2001 [17,26,27,28,29]. In our study, we recorded a prevalence of moderate inflammation in the PCa + PI group. In our series, a statistically significant association between moderate intraprostatic inflammation, especially inside the periglandular part of the prostate, was recorded in the PCa group. As such, we relied on the assessment of chronic inflammation according to a score summarizing precise parameters: anatomical location, grade of inflammation, and extent of inflammatory infiltrates. In this clinical scenario, future studies should: (a) identify different stimuli that may induce chronic prostatic inflammation and would lead to tissue damage; (b) identify the predictors of both chronic inflammation and PCa progression, and detect patients with chronic inflammation, which would be crucial to preventing PCa progression; (c) explore whether different grades and anatomic locations of inflammation might have a role in the development as well as the aggressiveness of prostate tumors; and (d) find patients with PI and, according to location and grade of inflammation, to develop target therapies to prevent PCa and its progression.

Our study has several limitations. First, the retrospective nature of our study. Second, although each core was evaluated by blinded dedicated pathologists with well-defined pathological criteria, the topographic location was designed according to arbitrary criteria, yet there are currently no validated frameworks to assess the location and grade of inflammation. Third, the analysis was performed on the anatomopathological evaluation by eight pathologists, limiting the reproducibility of our findings. For instance, our findings cannot be applied to patients with clinically suspected acute prostatic inflammation, repeat biopsy, active surveillance, previous BPH surgery, medical treatment with 5α- reductase inhibitors or saturation biopsies. As such, further studies are needed to confirm our findings in patients with these characteristics. Fifth, we could not identify the type of inflammatory infiltrate in the cores with PCa. Lastly, despite the study including a large number of patients, a relatively low number of cases with PCa and PI were included, and selection bias cannot be entirely ruled out.

## 5. Conclusions

PCa is a variable disease with a wide variety of pathological structures and different clinical courses. In our experience, prostatic inflammation is associated with the presence of PCa on biopsy cores. Well-designed prospective future studies are mandatory to confirm our findings and assess whether a structured identification and reporting of the presence and distribution of PI on biopsy cores might be used alone or in combination to identify patients that might benefit from target therapies to prevent PCa occurrence and/or progression.

## 6. Pros-IT2 Study Group

Brunelli Matteo (Verona, AOU Integrata Verona); Fiorentino Michelangelo (Bologna, Policlinico Sant’Orsola-Malpighi); Franzi Francesca (Varese, Ospedale di Circolo e Fondazione Macchi); Iaria Loredana (Padova, Policlinico di Abano Terme); Nesi Gabriella (Firenze, AOUC Careggi); Prosperi Enrico (Perugia, AO Perugia); Santi Raffaella (Firenze, AOUC Careggi); Sidoni Angelo (Perugia, AO Perugia); Toncini Carlo (Genova, Azienda Ospedaliero-Universitaria San Martino); Vecchione Andrea (Roma, A.O. Sant’Andrea).

## Figures and Tables

**Figure 1 cancers-15-00630-f001:**
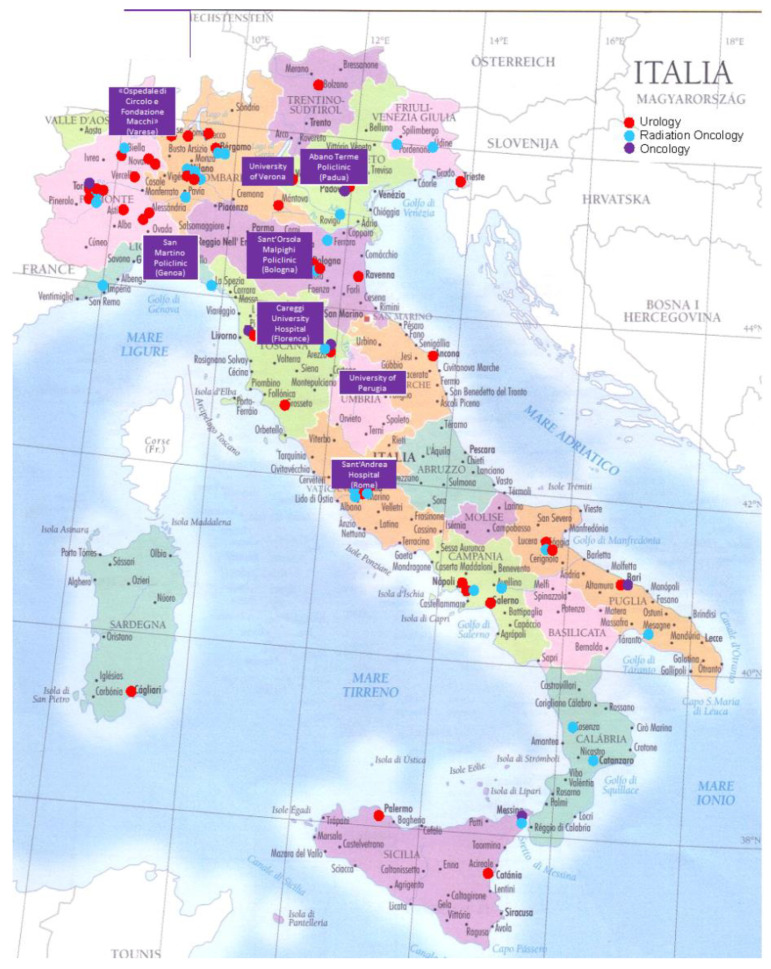
Centers participating in the Pros-IT CNR study (red, blue or purple dots) and centers participating in the Pros-IT2 study (purple rectangles).

**Table 1 cancers-15-00630-t001:** General characteristics of the patients included in the Pros-IT2 study.

	Patients (*n* = 197)
Age at diagnosis, years, mean ± SD	65.5 ± 7.5
Education > lower secondary school, *n* (%)	154 (78.6)
BMI, kg/m^2^, mean ± SD	26.7 ± 3.2
Smoking status, current smoker, *n* (%)	28 (14.3)
Diabetes mellitus, *n* (%)	20 (10.2)
Family history of prostate cancer, *n* (%)	48 (25.0)
Family history of breast cancer, *n* (%)	12 (7.9)
Family history of ovarian cancer, *n* (%)	4 (2.8)
PSA at the diagnosis, ng/mL, median (Q1, Q3)	6.7 (4.9, 9.8)
Clinical T staging, *n* (%)	
T1	123 (63.1)
T2	65 (33.3)
T3, T4	7 (3.6)
Gleason Score at the diagnosis, *n* (%)	
≤6	94 (47.7)
3 + 4	53 (26.9)
4 + 3	15 (7.6)
≥8	35 (17.8)

**Table 2 cancers-15-00630-t002:** Characteristics of the cores with neoplasia (*n* = 749).

	*n* = 749
Number of cores for each patient, mean ± SD	11.3 ± 4.9
Positive cores for each patient, *n*, mean ± SD	4.1 ± 2.9
Grade group, *n* (%)	
1	276 (36.9)
2	203 (27.1)
3	81 (10.8)
4	128 (17.1)
5	61 (8.1)
Gleason score, *n* (%)	
3 + 3	280 (37.4)
3 + 4	199 (26.6)
3 + 5	2 (0.3)
4 + 3	81 (10.8)
4 + 4	80 (10.7)
4 + 5	57 (7.6)
5 + 3	5 (0.7)
5 + 4	21 (2.8)
5 + 5	24 (3.2)
Risk, *n* (%)	
Low	280 (37.4)
Intermediate	280 (37.4)
High	189 (25.2)

**Table 3 cancers-15-00630-t003:** The relationship between inflammation and neoplasm.

	Neoplasia	No Neoplasia	*p*-Value
	(*n* = 749)	(*n* = 1471)	
Anatomical location, *n* (%)			0.0031
Stromal	351 (55.4)	735 (63.5)	
Periglandular	244 (38.5)	360 (31.1)	
Glandular	39 (6.2)	62 (5.4)	
Grade of inflammation, *n* (%)			0.0004
Mild	431 (68.0)	884 (76.4)	
Moderate	187 (29.5)	246 (21.3)	
Severe	16 (2.5)	27 (2.3)	
Extension of inflammatory infiltrates, *n* (%)			0.0131
Focal (<10%)	486 (76.8)	936 (80.9)	
Multifocal (10–50%)	139 (22.0)	195 (16.9)	
Diffuse (>50%)	8 (1.3)	26 (2.3)	
Inflammatory score, mean ± SD	4.1 ± 1.3	3.9 ± 1.2	0.0001
Glandular Disruption, *n* (%)	24 (3.5)	27 (2.2)	0.0821
Corpora Amilacea, *n* (%)	210 (31.0)	339 (27.5)	0.1054
Chronic inflammation, *n* (%)	634 (84.7)	1157 (78.7)	0.0007

## Data Availability

The data presented in this study are available on request from the corresponding author (Prof. Mauro Gacci).

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
