# Peer review of "Inflammation and Prostate Cancer: Pathological Analysis from Pros-IT CNR 2"

_cancers, 2023, doi:10.3390/cancers15030630_

Round 1

Reviewer 1 Report

The authors present results from re-evaluation of more than 2000 biopsy cores and associate the presence of CaP with chronic inflammation.

The work is ambitious and seemingly well executed. The hypothesis is interesting and of potential importance.

Minor comments

1. Line 143. A logistic regression model (outcome is presence of Cap) is promised in the method section but no signs of corresponding ORs in the result-section.

2. Line 109. Delete "abnormal"

Author Response

Risposta reviewer 2

Dear Reviewer, we would like to thank you for spending your time on reviewing our work and for the advices that were given. Below you may find the corrections that have been made based on your suggestions.

  1. Since this type of analysis has not been performed, we deleted the paragraph on Method section.
  2. We deleted abnormal from line 109.

Reviewer 2 Report

The authors present a manuscript in which they associate prostatic inflammation with prostate cancer (PCa). In fact they found an association with PI and the detection of prostate cancer. 

The question of course is, what was first. Inflammation or prostate cancer? Was the inflammation first and then PCa arose, or vice versa, did the PCa cause inflammation. Can the authors say something about this?

 What are the clinical consequences of these findings? The diagnosis of inflammation is only made when biopsies already have been taken. Most important, is PCa combined with PCa associated with a worse prognosis or with a better prognosis? Can the authors say something on the association between PCa with and without inflammation with the risk of biochemical relapse after treatment? Or with metastases on staging bone scintigraphy or PSMA PET/CT.

These questions make the paper more worthwhile and clinically useful

Author Response

Risposta reviewer 2

Dear Reviewer, Thank you for the time you have dedicated to our work. Thanks for the comments which have undoubtedly given us excellent insights that could improve the clinical impact of our work.

1.Data on prostatic inflammation and prostate cancer available were cross-sectional and not longitudinal; for this reason, it was not possible to define the possible cause and the possible effect, but it was only possible to evaluate their pathological association. We have now stressed this aspect in the Discussion, among limitation of the study, also stressing the need for specific longitudinal studies. 

  1. Unfortunately, bone scintigraphy and PET/CT were not available for the present study. In relation to the risk of biochemical and/or clinical relapse during follow-ups, we added the following paragraph (3.4 Follow up and risk of BCR): considering Pros-IT patients for whom assessment on cores and inflammation were available (n=197), no significant association between inflammation and relapses was found (16.7% among patients without inflammation vs 14.2% among patients with inflammation; p=0.7281). The percentage of patients with PSA>0.07 ng/ml at the 12 months follow-up was 12.1% among those with inflammation, and 8.3% among those without inflammation (p=0.7017), and at the 24 months follow-up 13.2% and 0.0%, respectively (p=0.3847).

Round 2

Reviewer 2 Report

The authors have responded well to the comments. I would suggest that the authors add a final line to the conclusion section of the manuscript.

We found an association between prostate inflammation and the presence of prostate cancer, but for now, patients with concurrent prostatic inflammation do not seem to have a worse prognosis. 

Author Response

Dear Reviewer, Thanks once again for your feedback. As requested, we have included this comment in the conclusions paragraph.